# Written Informed Consent—Translating into Plain Language. A Pilot Study

**DOI:** 10.3390/healthcare9020232

**Published:** 2021-02-20

**Authors:** Agnieszka Zimmermann, Anna Pilarska, Aleksandra Gaworska-Krzemińska, Jerzy Jankau, Marsha N. Cohen

**Affiliations:** 1Department of Medical and Pharmacy Law, Faculty of Health Sciences, Medical University of Gdańsk, 80-210 Gdańsk, Poland; anna.pilarska@gumed.edu.pl; 2Department of Nursing Management, Institute of Nursing and Midwifery, Medical University of Gdańsk, 80-211 Gdańsk, Poland; aleksandra.gaworska-krzeminska@gumed.edu.pl; 3Department of Plastic Surgery, Faculty of Medicine, Medical University of Gdańsk, 80-214 Gdańsk, Poland; jerzy.jankau@gumed.edu.pl; 4College of the Law, University of California Hastings, San Francisco, CA 94102, USA; cohenm@uchastings.edu

**Keywords:** informed consent, patient’s rights, plain language, plastic surgery, work environment, quality management practice, risk management

## Abstract

Background: Informed consent is important in clinical practice, as a person’s written consent is required prior to many medical interventions. Many informed consent forms fail to communicate simply and clearly. The aim of our study was to create an easy-to-understand form. Methods: Our assessment of a Polish-language plastic surgery informed consent form used the Polish-language comprehension analysis program (jasnopis.pl, SWPS University) to assess the readability of texts written for people of various education levels; and this enabled us to modify the form by shortening sentences and simplifying words. The form was re-assessed with the same software and subsequently given to 160 adult volunteers to assess the revised form’s degree of difficulty or readability. Results: The first software analysis found the language was suitable for people with a university degree or higher education, and after revision and re-assessment became suitable for persons with 4–6 years of primary school education and above. Most study participants also assessed the form as completely comprehensible. Conclusions: There are significant benefits possible for patients and practitioners by improving the comprehensibility of written informed consent forms.

## 1. Introduction

The informed consent (IC) process is an important aspect in clinical practice as it is how persons seeking medical treatment, or their surrogate decisionmakers, indicate their understanding of a proposed medical procedure and their agreement to proceed [1]. An informed consent form is a critical element in the IC process of sharing information and communicating decisions between medical practitioners and patients. Informed consent also has an ethical dimension that is codified in legal statutes and institutional regulations that require physicians to obtain informed consent prior to treatment [2]. Courts almost unanimously treat the lack of IC as an instance of medical negligence. Consequently, a physician must assess a person’s ability to understand the relevant medical information and the implications of treatment alternatives, and whether they are able to make an independent, voluntary decision. Further, the medical records must contain evidence of the consultation and the person’s decision, and a copy of the signed informed consent document [3].

The purpose of an IC form is to provide relevant and necessary information in a manner that can be easily understood by the patient. Patients who receive inconclusive information are likely to misunderstand the substance and import of the information they need to fully grasp before giving their consent. In Polish jurisdictions, written IC is required, though there are some rare exceptions. Requiring that a patient signs an IC form ensures that there is documentary evidence that the patient has in fact received the necessary information and subsequently granted their IC for the procedure. Informed consent is also predicated on the patient being given sufficient time and opportunity to read, evaluate, and consider the information prior to their consent being required. Furthermore, the information, and the patient’s understanding of it, must be sufficient for the patient to be able to make an autonomous decision about consent [4]. Securing a patient’s consent is a process requiring effective communication between the physician, the whole medical team, and the patient. Where relevant, the discussion between a patient and the health care provider should allow the patient’s caregiver to have questions considered and answered.

While in Poland there is no comprehensive set of national standards or guidelines regulating the medical professional regarding IC requirements and processes, de facto guidelines do exist across various laws of parliament and institutional codes of ethics. These make it clear that medical professionals should provide comprehensive information about the proposed treatment, including alternatives to the proposed treatment, and ought to disclose any risks to the patient, thus enabling patients to make knowledgeable decisions about their medical care [5]. Polish codes also make it clear that consent must be given voluntarily, and that patients must have the freedom to revoke their consent. Poland’s de facto guidelines also state that information disclosed should include the following:•Condition/disorder/disease that the patient has/suffers from;•Necessity for further testing, if any;•Natural course of the condition and possible complications;•Consequences of non-treatment;•Treatment options available;•Potential risks and benefits of treatment options;•Duration and approximate cost of treatment;•Expected outcome;•Follow-up required [6].

A critical IC issue is the degree of clarity of the information provided and the risks associated with any lack of clarity or patient understanding. In the process of obtaining an IC, the medical professional must provide information in a manner that is clear and intelligible to a competent person so that they may choose to accept or refuse treatment [7]. As stated by the American College of Surgeons, “patients should understand the indications for the operation, the risks involved and the result that it is hoped to attain” [8]. Even when a treatment is elective, such as in many cases of cosmetic surgery, a person’s request for a non-therapeutic treatment cannot be considered as a surrogate for consent, and surgery carried out without a patient’s informed consent remains illegal. Thus, written informed consent is vital and requires that the person seeking treatment must receive and understand all relevant information and documentation [9]. A significant risk, when the IC process is incomplete or inadequate, is litigation. While a physician is not responsible for non-negligent adverse outcomes in cases of therapeutic treatments, in cases of elective plastic surgery, questions of whether planned outcomes were achieved or even achievable can sometimes be a subjective judgement, and the rate of legal claims in these cases is high [10,11]. The existence of a signed IC form will not always preclude malpractice claims by the patient. However, a properly conducted informed consent process is an important mitigator of the risk of malpractice suits, as studies show that a poor IC process is one of the most common causes of malpractice litigation when a procedure fails or results in complications [12].

The degree of a person’s understanding when receiving IC information is central to the effectiveness of the IC process. Many factors influence a patient’s ability to understand the content of informed consent information, and written informed consent is preferable where significant risk is involved. Written consent forms are common in Poland, but they are often generic documents, prepared by lawyers with blank spaces left for specific information about the planned procedure to be added by the physician, and the language used is hard to understand. While standardised consent forms can potentially improve the efficiency of the consent process, as well as providing security for the surgeon and the patient that all aspects are included, we believe they are not always easily understood; and in Poland, there is an absence of established norms, standards, or guidelines for the use of plain language in these forms. International studies indicate that when written information is provided to patients as part of the IC process, it should be fully understandable to the patient, regardless of their degree of literacy, but that IC forms often over-estimate people’s literacy levels. For example, United Kingdom data show that the general literacy level of one in every six people is lower than that expected of an 11-year-old school student [13]. Similarly, data from the USA show that one in every four American adults have low literacy skills, with national surveys estimating that there are 40 million functionally illiterate and 50 million marginally literate adults [14]. Given that the average reading skill level of US adults is 8th-grade level, the American Medical Association (AMA) recommends that written information for patients should not exceed a 6th-grade reading level [15]. While most adults read at an 8th-grade level, 20% of the US population reads at or below a 5th-grade level. However, most healthcare materials are written at a 10th-grade level. Older patients often face additional challenges because their reading and comprehension abilities are also influenced by their cognition, vision, and hearing status [16]. Plain language consent forms have been found to be easy to read and understand, and Spellecy at al., concluded that if an information form is easier to read, patients feel less anxious and experience greater satisfaction from participating in a therapeutic process [17].

The link between patient comprehension and the readability of the IC information is central to our study. It is known that a person’s ability to understand the written information they are given can be significantly improved if the readability of the text is adapted to their reading level. Therefore, it is essential that IC forms are written in clear, simple, and direct language to ensure readability. Words longer than three syllables, long sentences, passive sentences, and medical vocabulary are among many factors reducing the readability of patient information in the IC process [18]. Also, the level of patients’ comprehension declines proportionally to the increasing length of the informed consent statement. It has also been shown that IC forms written in the first person, using “I / me” sentences, or in the second person, using “you/your” sentences, make the text more user-friendly and easier to understand. Studies have identified four types of intervention to improve patient comprehension of IC forms: (1) additional written information, (2) audio-visual/multimedia interventions, (3) extended informed consent discussions, and (4) test/feedback techniques [19]. All the published studies we reviewed were of assessments of English language IC processes and forms. There are no corresponding studies of Polish-language material. Therefore, our research has focused on the readability of Polish-language IC forms, specifically, and arbitrarily, a single generic IC form used in a plastic surgery setting. 

The aim of our study was to be able to create a clear, easy-to-understand IC form and that will be easy to understand as a step towards establishing a model of the use of plain language in the IC process. To that end, our study includes the rewriting of a standard IC form. A consequence for our study could be the introduction of sustainable changes to informed consent processes in Poland, leading to an enhanced work environment for medical practitioners, and improved patient outcomes.

## 2. Materials and Methods 

### 2.1. Study Design and Data Collecting Process

The research was divided into two phases. In the first phase a standard, 3-part informed consent form for reconstructive and plastic surgery currently used in a teaching hospital in the north of Poland was analysed. The form chosen for the study was the result of the researchers’ prior consultations with the Polish Society of Plastic, Reconstructive and Aesthetic Surgery. A linguistic analysis software packaged was used. The analyzed form had three sections. The first section was for recording the hospital name and address, and the name of the doctor providing the consent information. The second section of the form was for clinical information on the planned procedure, and information on possible risks and complications. The third section was for the consent declaration and signatures. Our analysis only evaluated the second section of the form. The preliminary analysis used a recognized Polish text comprehension analysis program (jasnopis.pl, SWPS University, Warsaw, Poland) to evaluate whether the document was grammatically correct, concise, and easy to understand. The software was developed and verified in a study that involved 1000 individuals whose native language was Polish, using 35 different statements, each with a different level of difficulty [20]. The Jasnopis software has been used previously in several Polish studies [21,22,23]. The software makes a qualitative evaluation of each sentence on a 7-point scale:1st–3rd years of primary school level—Child’s play4th–6th years of primary school level—Very easyJunior secondary school level—EasySecondary education level—A little difficultCollege (bachelor/engineer) level—Moderately difficultUniversity (masters) level—DifficultDoctoral studies (PhD) or specialist knowledge level—Complicated and professional.

The “Jasnopis” tool is for measuring the “fogginess” of Polish language texts. It is based on Robert Gunning’s readability formula. The original Gunning formula, designed to measure the readability of English texts, had to be adapted to the Polish language, which differs from English in, among other things, its inflections, and the average length of words. In the English language, words which have three or more syllables are assumed to be difficult; in the Polish language, where words are on average longer, those having four or more syllables are the equivalent of three-or-more-syllable words in English [24]. In addition to assessing a text against the 7-point scale cited above, the “Jasnopis” program identifies “difficult” words (defined as having four or more syllables, excluding those words considered to be generally known), over-long sentences, over-difficult paragraphs; and, in addition to a statistical report and graphs to represent the degree of lexical similarity and difference from one paragraph to another, the program’s analysis suggests possible substitutions for difficult words (synonyms, hyponyms, hypernyms); and reports of stylistic consistency, or otherwise, across the whole text and in relation to the user-nominated desired style [25]. “Jasnopis” was developed by the Polish National Center of Science Foundation and has been endorsed by the Polish Minister of Science [26]. 

The Jasnopis analysis of the original IC form that we used in our study suggested replacing several difficult or words or medical jargon with less complicated synonyms, as well as suggesting simplifications of the sentence structure. Therefore, we re-wrote long sentences as several shorter ones, and difficult phrases were replaced with simpler forms. For example, “alternative methods” was replaced with “other possible methods, “prognosis” with “foreseen results”, and “complications” with “problems.” Our aim was to create a revised IC form that would rank in a range between 2 (“very-simple-to-read”) and 3 (“simple-to-read”). Following our revisions, which conformed to the Jasnopis recommendations, we believed the content of the form would be more accessible to the reader. 

In the second stage of the research project, we enrolled 160 volunteers from different age groups, with a range of education levels to test the modified IC form and identify too-complicated language. The study group comprised lay persons (i.e., non-healthcare professionals). This stage of the project was conducted in an academic surgical unit of large teaching hospital, using a group-administered questionnaire. Participant criteria included the following: volunteers were patients of the surgical unit, had never been employed in healthcare services or worked with medical documentation, were adults, and fluent in spoken and written Polish. Exclusion criteria: incapacitated persons, persons under 18, persons writing or using official documents in their daily work (such as in local and central government administration roles). Participation in the research was voluntary and conditional on obtaining each volunteer’s consent. Before participating, volunteers were given written and oral information by researchers that described the purpose of the research and the method to be used in conducting it, and instructions for participation. Our data collection methods guaranteed complete anonymity. For the purposes of this stage of the research, the text of the modified IC form was divided into 14 parts. Volunteers received the entire form and were asked to match each sentence with one of three evaluative descriptors: “I fully understand it, I do not need any more information”, “I partly understand it and I need more information,” or “I don’t understand it and need to talk to the nurse or doctor.” The response forms were distributed and collected anonymously by one of the researchers (a PhD student), during May 2019. Thus, while the first stage of our study was to achieve an easy-to-understand revision of the original IC form, the second stage of the study was to verify, using participants’ assessments, whether the simplified text was in fact easy to understand. The project was developed as a first exploratory study for future extended research.

The collected data were digitized manually, and a unique number was assigned to each questionnaire by the person responsible for entering the data into an Excel database. Data coding and entry accuracy were verified by a second person from the research team. Non-response items were excluded from the analysis. All statistical calculations were carried out using STATISTICA version 12.0 (StatSoft. Inc, Tulsa, OK, USA, 2014, www.statsoft.com (accessed on 19 December 2020)) and an Excel spreadsheet. Pearson’s and/or Spearman’s correlation coefficients were used to verify the existence, strength, and direction of relationships between variables of age, and educational level. The level of significance in all calculations was assumed to be *p* < 0.05.

### 2.2. Ethical Considerations

The research project received approval from a Medical University Independent Bioethics Committee for Scientific Research and included the finding that the project constituted “non-invasive research”. All participants gave their informed and voluntary consent prior to participation. The study was conducted in accordance with the requirements of Poland’s data protection legislation.

## 3. Results

### 3.1. Readability Test Using Linguistic Software

Jasnopis analysis of the original IC form suggested revisions were needed. The various revisions of the form to reduce its linguistic complexity improved its accessibility, as indicated by the results of the second Jasnopis analysis. As Table 1 shows, after the revisions, the level of difficulty according to the Jasnopis software fell from levels 5, 6, and 7 (moderately difficult, difficult, and professional) to a combination of 2 (very-easy-to-read), 3 (easy-to-read), and 4 (a little difficult-to-read). In 5 of the 14 segments that we had divided our text into, for assessment purposes, simplified vocabulary replaced professional and technical medical language. In five segments, whole sentences were simplified. Five long sentences were re-written as shorter ones. Four sentences were shortened. In three segments, sentences were personalised using the word “my.” Overall, only three sentences remained unchanged. In two segments, complicated vocabulary was omitted. As a result of all these edits, Jasnopis assessed the sentences as either easy- or very-easy-to-read. Only one sentence remained with a “moderately difficult” readability classification, due to that fact that it would be difficult to replace “diagnosis” and “complications” with meaningful simpler synonyms.

### 3.2. Readability Analysis in Study Group

In the second stage of the project, involving volunteers, 74 men and 86 women participated. All participants were adults (18+). We did not apply an upper age limit for respondents. Participants had a range of educational levels, having either completed 6 years of primary schooling or graduated from secondary school, or University (with a bachelor’s or master’s degree or PhD). Respondents’ demographic data are presented in Table 2. Those respondents who had completed primary school were from the 40–60 (8.1%) and over 60 (5.0%) age groups. The post-secondary school respondents were mostly in the 40–60 age bracket (28.0%) and those with university degrees were from the 18–39 (13.8%) and 40–60 (13.8%) age groups.

Table 3 presents data on the participants’ assessment of the readability of the informed consent form. Volunteers were asked to assess each of the form’s 14 segments separately. Our findings indicated that the IC, after the earlier modifications, was entirely understandable for 78% of respondents. We found a strong correlation between understanding and young age, where the younger group (aged up to 39 years) were significantly more likely to understand the entire IC form (Chi^2^ = 23.87 *p* = 0.0001). There was also a significant correlation between University-level educational and full understanding (Chi^2^ = 12.09, *p* = 0.0024) and whereas we found weaker and not statistically significant correlations in relation to the other educational levels.

## 4. Discussion

While the initial stage in the preparation of an informed consent form must focus on its required content, that stage should be followed by adopting language standards that guarantee that the text is easy-to-read and understand by patients. Our study used a single template-type form, and while it had all the required content, our two-stage assessment of its readability revealed that the language required substantive revision to make it easy-to-understand—a finding that was consistent with the literature. However, the novel Polish-language focus of our study meant that we were only able to cross-check our findings with studies that had assessed English-language issues of readability and comprehension. One factor we were concerned with was how to assess the readability of the language of an IC form. Studies that concern readability of consent forms created in English usually use the Flesch–Kincaid Grade Level scale for their text assessments, which uses an index value for text that corresponds to the education level of readers who will be able to understand the text [27,28,29]. A second factor our study was concern with was the possible reading levels of persons reading the IC form. One English study estimated that 50% of the population possesses a reading level below 8 points on the Flesch–Kincaid Grade Level scale, meaning a relatively simple text suitable for students less than 13 years of age. Extrapolating from this, the authors of that study suggested that a suitable readability level for all documents intended for patients would be 4–6 points (a simple-to-very-simple text) [28]. Another readability scoring system reported in the literature is the FRES index (Flesch Reading Ease Score). One study used the FRES index to assess the readability of information about 32 procedures approved by the British Orthopaedic Society. Using volunteer participants, the study generated an average index value of 63.6, which corresponds to a 13–15-years-old readability level. On this basis it was determined that only 43% of the English population would be able to understand the assessed information material. To remedy this, the study authors recommended that IC documents should be revised to meet the reading ability level of 11-year-old children, or a score of 90–100 on the FRES index [30]. Another study, which assessed the readability of IC forms for invasive procedures from all the surgical inpatient hospitals in the State of Rhode Island, used a range of indices: Flesch Reading Ease Formula, Flesch–Kincaid Grade Level, Fog Scale, SMOG Index, Coleman–Liau Index, Automated Readability Index, and Linsear Write Formula. From the resulting readability scores, the authors calculated a composite Text Readability Consensus Grade Level. The assessment showed that on average, the IC forms had a readability level suited to 15th-grade level (i.e., the third year of college in the US, or university in the UK), which is significantly higher than the average US adult at 8th-grade reading level, making comprehension difficult-to-impossible for many persons [31]. This finding was like our own. We found that 11 out of 14 parts of the form we analysed scored at 4–7 on the Jasnopis scale (difficult-to-complicated). Thus, most of the text, which had been drafted by lawyers, was only comprehensible to persons with University-level education, and some parts of the text were only comprehensible to those with University-level medical learning. These results and the evidence of other studies indicated that modifications and simplifications were required if the IC form was to be more widely understood by patients [27,28,30].

In a Croatian IC information readability study, the findings included the recommendation that the use of simple linguistic phrases, short sentences, readable subtitles, and instructions that avoid medical terminology would enable patients to better understand the content of these forms [27]. A USA study of patient information leaflets from a family medicine clinic made a comparative readability assessment, scoring the same pamphlet both before and after removing medical terminology [32]. Results of that study indicated that the reading levels for all brochures were significantly lower after the removal of medical terminology, but that they remained above the 5th-to-6th-grade level recommended by health education experts. These findings have implications for healthcare professionals in relation to the development and evaluation of patient education materials. As a solution, removal of all medical terminology or replacement with simple words, may not always be either practical or realistic. Studies have shown that any medical terminology that is essential for conveying the appropriate information, which therefore must be retained in the patient information, should be supported by coherent and readable definitions [33]. Consistent with these recommendations and our own findings, we made several vocabulary modifications to the IC form we were analyzing, including replacing technical and specialist words with plain language forms that would be familiar to people who did not possess specialised health knowledge, and replacing difficult-to-understand words with simpler ones in 50% of the text. Regarding the reported issues with medical jargon in IC forms, we retained some technical and specialist vocabulary for practical reasons, however, we also provided easy-to-understand definitions for each of these terms (e.g., “therapy using drugs” for pharmacotherapy, and “blood products or drugs made from blood” for blood-based preparations).

Sentence structure is also critical for easy of understanding. Studies have shown that breaking longer sentences that contain several ideas into shorter sentences that contain a single idea has comprehensibility benefits [34]. Sentences should be short, simple, and direct [35]. In our study, seven segments (50% of the IC form) contained long sentences that we divided into shorter ones. Therefore, in our study we both shortened sentences by dividing too-long sentences, and simplified those that were too complex, in an editing process that improved the readability of the IC form making it “easy-to-read” on the Jasnopis scoring index.

Finally, the way the information addresses the reader has been found to be a significant factor in readability scoring, specifically in relation to the use of pronouns. Studies have reported that IC forms should be written using active verbs and worded as if the medical professional is speaking directly to the patient, as supported in the literature [35]. We applied this recommendation throughout our IC form, thus giving preference to addressing patients with a conversational tone in the 2nd person, rather than using the impersonal 3rd person with its associated passive grammatical construction.

Having made all the language revisions mentioned above in the first phase of our study, and re-assessed the text using the Jasnopis scoring index, which generated an “easy-to-read” score for the revised text, we instructed our volunteer participants to assess the IC form. Most volunteers found that the modified informed consent form was understandable. The results of our study indicate that documents prepared by experienced lawyers, though consistent with Polish legal requirements, may not be fully comprehensible for patients without language revisions aimed at ease of readability. It is reasonable to believe that this conclusion would also apply to documents in other languages, such as in English. Patient understanding of the health information they receive is central to the validity of the whole informed consent processes. However, during the process, patient comprehension of written material is frequently overestimated [36]. Crepeau et al., found that patients in a surgical setting performed at unexpectedly low levels when their comprehension and recall was assessed immediately after a detailed consent form briefing [37]. Such evidence suggests that consent forms may be ineffective with significant numbers of patients because they cannot sufficiently comprehend the content [31]. The proven and significant correlations between health literacy and health outcomes, and the associated correlation between reading ability and health literacy, has led the US Department of Health and Human Services (USDHHS) to recommend a 6th-grade reading level (equivalent to a UK reading age of 11–12 years) for all patient-facing health literature [30]. Poland has not yet created any such guidelines.

The limitation of our study was the small sample size; however, the research was undertaken as a pilot study to produce preliminary, rather than generalized data. In addition, we did not involve human readers to test the original text prior to analyzing it with the linguistic software. Nor did we compare the readability and comprehension scores of our human readers with the software scores before and/or after the text modifications, which would have provided an additional means of demonstrating improved readability. Our decision to analyze a single IC form in the largely unregulated situation of Poland, was based in part on discussions with the scientific society we consulted with, and on our study aim to achieve an IC form that we could recommend as a benchmark for future use.

## 5. Conclusions

While the initial stage in the preparation of an informed consent form must focus on its required content, that stage should be followed by adopting language standards that guarantee that the text is easy-to-read and comprehensible by patients. This should involve ensuring the use of clear section headings, the use of personal pronouns, short words, words with few syllables, short sentences (including dividing long sentences into several shorter ones), plain language substitutes for medical jargon, and—where specialist and technical terminology must be retained for practical reasons—ensuring each term is supported by readable definitions. Such texts should also be tested with a recognised evaluation tool to verify their readability. This process for creating an informed consent form (draft, revise, and test) would enable healthcare centres to facilitate improved cooperation and outcomes for healthcare personnel and patients. Such processes for preparing easy to understand informed consent forms for patients in surgical and other medical settings could lead to improved patient understanding medical objectives and the potential outcomes of proposed therapies.

## Figures and Tables

**Table 1 healthcare-09-00232-t001:** Results of analysing the informed consent form with the Jasnopis application.

Sentence Number and Subject Matter	Difficulty of the Text According to the Application	Changes Introduced	Assessment According to the Application after Modification
1. information about the patient’s state of health	6—difficult	− dividing long sentences into shorter ones − replacing more difficult words with simpler ones, − using personal pronouns	3—easy
2. information about the diagnosis	5—moderately difficult	− dividing long sentences into shorter ones, − replacing more difficult words with simpler ones	3—easy
3. information about drugs given	5—moderately difficult	− a simplification of sentences, − using personal pronouns − medical terms supported by easy-to-read definitions	3—easy
4. information about the nature and purpose of the surgery	3—easy	− -no revisions	3—easy
5. information about alternative methods or no alternative methods	7—complicated	− dividing a long sentence into shorter ones, − replacing more difficult words with simpler ones	4—moderately difficult
6. information about the assumed therapeutic effect with the stipulation that the result of operation is not certain	7—complicated	− dividing long sentences into shorter ones, − replacing more difficult words with simpler ones, − a simplification of sentences	3—easy
7. information about the risks (also about the risks connected with the substances to be used)	5—moderately difficult	− dividing long sentences into shorter ones	3—easy
8. information about each typical potential complication	5—moderatelydifficult	− replacing more difficult words with simpler ones, − using personal pronouns	3—easy
9. information about frequent ailments and side effects	2—very easy	− no revisions	2—very easy text
10. information about the anesthetics that will be used (including allergic reactions)	2—very easy	− no revisions	2—very easy text
11. information about the possible necessity of transfusing blood or blood-based preparations	5—moderately difficult	− dividing long sentences into shorter ones, − replacing more difficult words with simpler ones, − medical terms supported by easy-to-read definitions	3—easy
12. declaration of competence to sign	7—complicated	− dividing long sentences into shorter ones, − replacing more difficult words with simpler ones, − a simplification of sentences	3—easy
13. declaration of voluntary decision	6—difficult	− a simplification of sentences	3—easy
14. declaration of state of mind (the influence of any doping)	6—difficult	− a simplification of sentences	3—easy
**Mean:**	**5—moderately difficult**		**Mean: 3—easy**

**Table 2 healthcare-09-00232-t002:** Respondent characteristics: age, gender, and educational levels.

Study Group (*N* = 160)		
Parameter	*n*	%
**Age**		
18–39	54	33.8
40–60	80	50.0
over 60	26	16.2
**Gender**		
Female	86	53.8
Male	74	46.2
**Educational level**		
Primary school (completed)	27	16.8
Secondary school (completed)	81	50.6
University degree	52	32.6

**Table 3 healthcare-09-00232-t003:** Results of testing the informed consent (IC) form with participants.

Form Segment	Percentage of Respondents for Whom the Fragment of the Text Was:		
	Entirely Understandable (%)	Party Understandable (%)	Incomprehensible (%)
1.	69	22	9
2.	71	21	8
3.	81	13	6
4.	94	6	0
5.	58	29	13
6.	71	24	5
7.	74	19	7
8.	78	13	9
9.	96	4	0
10.	95	5	0
11.	69	21	10
12.	76	19	5
13.	77	19	4
14.	88	11	1
Median	78.0%	16.1%	5.9%

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
