# Peer review of "Written Informed Consent—Translating into Plain Language. A Pilot Study"

_healthcare, 2021, doi:10.3390/healthcare9020232_

Round 1

Reviewer 1 Report

This is a very well designed and well presented study on an important topic. All concerns have been addressed in the revisions.

Author Response

Thank You very much for Your opinion.

Reviewer 2 Report

Dear Authors, the paper is a qualitative study on Informed Consent, and the necessity of improving the readibility of such a form. The study is of interest, but I have some comments and suggestion for improving the manuscript.

Abstract:

-please, move the aim at the end of the background (as usually in scientific literature. Please, specificy the meaning of "health-literate"...I am not sure what you mean in this context (the same in the main text).

-the software you used should be cited in the abstract as well

Introduction:

  • It is too long and redundant, difficult in the readibility...please reduce it with a focus on the main topic.

Material and Method:

  • On p.4 line 154: the software should be fully described with references. The validation process in Poland of this software should be better described. The references you provide do not help to understand how this software has been developed and validated in a Polish language and population.

Discussion:

-a first sentence of synthesys of the main finding should be inserted at the beginning of the discussion (the content as described in the lines 263-291 makes sense only if described in the context of the findings of this study)

- The discussion is too long and difficult to read, please revise it to improve the readibility.

-On line 356: only among the shortcomings the Authors state that this study is a pilot one. This aspect should be introduced before in the text, may be in the title as well. It is really difficult to understand why the Authors did not make an evaluation of readibility of the IC before the software evaluation, this would be very interesting to know and the methodology would be stronger than the current.

-English should be revised throughout the paper.

Round 2

Reviewer 2 Report

The paper improved a lot after the revision.

An additional point: "There was also a significant correlation between University-level educational and full understanding" (line 234)...plase specify what happened in other educational levels.

-line 191: please correct the word repetition (study)

Author Response

Thank you for your help and all opinions.

An additional point: "There was also a significant correlation between University-level educational and full understanding" (line 234)...plase specify what happened in other educational levels.

There was also a significant correlation between University-level educational and full understanding (Chi2 = 12.09, p = 0.0024) and whereas we found weaker and not statistically significant correlations in relation to the other educational levels.

-line 191: please correct the word repetition (study)

Thank You, we have removed the repetition of the word.

An additional point: "There was also a significant correlation between University-level educational and full understanding" (line 234)...plase specify what happened in other educational levels.

There was also a significant correlation between University-level educational and full understanding (Chi2 = 12.09, p = 0.0024) and whereas we found weaker and not statistically significant correlations in relation to the other educational levels.

-line 191: please correct the word repetition (study)

Thank You, we have removed the repetition of the word.

An additional point: "There was also a significant correlation between University-level educational and full understanding" (line 234)...plase specify what happened in other educational levels.

There was also a significant correlation between University-level educational and full understanding (Chi2 = 12.09, p = 0.0024) and whereas we found weaker and not statistically significant correlations in relation to the other educational levels.

-line 191: please correct the word repetition (study)

Thank You, we have removed the repetition of the word.

An additional point: "There was also a significant correlation between University-level educational and full understanding" (line 234)...plase specify what happened in other educational levels.

There was also a significant correlation between University-level educational and full understanding (Chi2 = 12.09, p = 0.0024) and whereas we found weaker and not statistically significant correlations in relation to the other educational levels.

-line 191: please correct the word repetition (study)

Thank You, we have removed the repetition of the word.

An additional point: "There was also a significant correlation between University-level educational and full understanding" (line 234)...plase specify what happened in other educational levels.

There was also a significant correlation between University-level educational and full understanding (Chi2 = 12.09, p = 0.0024) and whereas we found weaker and not statistically significant correlations in relation to the other educational levels.

-line 191: please correct the word repetition (study)

Thank You, we have removed the repetition of the word.

An additional point: "There was also a significant correlation between University-level educational and full understanding" (line 234)...plase specify what happened in other educational levels.

There was also a significant correlation between University-level educational and full understanding (Chi2 = 12.09, p = 0.0024) and whereas we found weaker and not statistically significant correlations in relation to the other educational levels.

-line 191: please correct the word repetition (study)

Thank You, we have removed the repetition of the word.

An additional point: "There was also a significant correlation between University-level educational and full understanding" (line 234)...plase specify what happened in other educational levels.

We added more details:

There was also a significant correlation between University-level educational and full understanding (Chi2 = 12.09, p = 0.0024) and whereas we found weaker and not statistically significant correlations in relation to the other educational levels.

-line 191: please correct the word repetition (study)

Thank You, we have removed the repetition of the word.

This manuscript is a resubmission of an earlier submission. The following is a list of the peer review reports and author responses from that submission.

Round 1

Reviewer 1 Report

The subject is interesting, but my main concern is about the number of CIs employed, just one! The analysis should be carried out with a larger number of CIs from different hospitals, with different wordings.

On the other hand, the text of the analyzed CI is not provided, so that its difficult to visualize its flaws, neither the result after the analysis and simplification of the language used in the CI after the modifications.

The data in table 2 are included in table 3, it should be rremoved and some data could be inserted in the text if authors consider it convenient.

The segments into which the IC is divided are not known, nor their expresion.

The discussion seems to be an introduction with a biliographic review on the subject and not a discussion, and many of the results are repeated.

Reviewer 2 Report

I teach and do research on biomedical ethics. And I can see just from the abstract that the authors of this article do not understand the fundamental ethical importance of informed consent. The abstract suggests that a written document is "informed consent." It is not. At best, a signed form merely documents that a valid informed consent conversation has taken place. It (the signed form) is neither necessary nor sufficient from an ethical perspective. Informed consent takes place in a conversation. A written document should never substitute for the conversation.

I would encourage you not to publish this article. Have the authors engage with an ethicist or take an ethics course before continuing with this line of research.

Reviewer 3 Report

This is a good contribution to the literature. The topic of health literacy and the related area of informed consent is an important one. The authors offer a solid review of the literature and place their study and its findings into the current body of knowledge. The focus on Polish language materials makes this a unique contribution. The methods seem sound, and the entire article is well-written and well-presented.

I only have some very minor comments and suggestions:

on page 3, line 103-104, the authors state that a poor IC process is one of the main reasons for medical litigation... that seems plausible but I think it would help to cite some evidence to support that claim.

I was wondering from the start why the researchers did not have human readers test the original form, before it was modified. I think it would have been interesting to compare readability and comprehension scores from the human readers as well as the software both pre and post modification. As it is, the authors can claim a certain level of readability of the new forms, but they can't necessarily say it is more readable than the previous one. I realize it is too late to change the approach, but maybe they could note that in the limitations.

The paper is very well written overall, but there are a few small typos and errors, so a copyedit is necessary.